# SurfCNN: A descriptor enhanced convolutional neural network

**Ahmed M.Elmoogy, Xiaodai Dong**∗ **& Tao Lu**∗,
Department of Electrical and Computer Engineering
University of Victoria
Victoria, BC, Canada
`ahmedelmoogy@uvic.ca,{xdong,taolu}@ece.uvic.ca`

**Robert Westendorp & Michael Xie**
Fortinet Inc.
Sunnyvale, CA, USA
`{rwestendorp,mxie}@fortinet.com`

## Abstract

Adding an image descriptor to the input significantly enhances the performance of a convolutional neural network. By incorporating a SURF descriptor for indoor localization applications, we report a simpler convolutional neural network with fast training speed and outstanding accuracy and without the need for a pretrained network in contrast to the state of art.

## 1 Introduction

Convolutional neural networks (CNN) are capable of extracting features from high dimension data such as images, videos, etc. that suits the application specific tasks. Such network, however, requires high dimensional optimization procedure in which the training time is significantly longer when the input dimension is large. On the other hand, image descriptors extract features from images through deterministic means that are orders of magnitude faster than CNN. The drawback of a descriptor is that, the output feature size is usually large compared to those from CNN as most of the image information is retained during extraction regardless of whether it is needed for the target application. Here, through the demonstration from an indoor localization application, we combine both technologies by first using an image descriptor to extract features from images. The feature set, which has significantly reduced dimension compared to the images, is input to a CNN to extract more useful features with further reduced dimension. The combined techniques result in a significant fewer parameters in the CNN and the training time is reduced accordingly.

In pre machine learning era, most image based localization tasks were done through place recognition or finding the position from 2D-3D correspondence. For example (Sattler et al., 2015) used image retrieval techniques to search for the similarity between the current image and images in the database. Consequently, the position at which current image was taken can be estimated. Other techniques (Sattler et al., 2011; 2017) rely on finding the 2D-3D correspondence via descriptor matching between the 2D image and 3D model using e.g., structure from motion (sfm) (Forsyth & Ponce, 2011). Both techniques, however, highly depend on handcrafted features that cannot be generalized in different environments.

There are various feature detectors and descriptors including Scale-Invariant Feature Transform (SIFT) (Lowe, 2004), Speeded Up Robust Feature (SURF) (Bay et al., 2008), Features from Accelerated Segment Test (FAST) (Viswanathan, 2009), Binary Robust Independent Elementary Features (BRIEF) (Calonder et al., 2010) and Oriented FAST and Rotated BRIEF (ORB) (Rublee et al., 2011). Among them, SURF is being used extensively in computer vision applications such as face recognition (Du et al., 2009), visual simultaneous localization and mapping (SLAM) (Engelhard et al., 2011) and object detection (Chincha & Tian, 2011). SURF relies on box filters and integral images which make feature extraction faster. (Bayraktar & Boyraz, 2017) demonstrated that by

using SURF descriptors the highest accuracy can be achieved in image matching for indoor localization. Typically, SURF can convert an image with around 1 million pixels into feature set with fewer than 20 thousand values, sorted by the corresponding Hessian threshold. That significantly reduces the data dimension without noticeable loss of image information.

With the booming of machine learning, image based localization using neural networks reaches high performance enhancement. For example, (Kendall et al., 2015) used transfer learning (Oquab et al., 2014) and the pretrained GoogLeNet (Szegedy et al., 2015) to regress the pose of the camera (position and orientation) with 0.44 m accuracy. Recently, (Hazirbas et al.) used similar architecture but with Long-Short Term Memory (LSTM) (Hochreiter & Schmidhuber, 1997) to memorize good features, leading to a better accuracy of 0.31 m. (Melekhov et al., 2017) used another pretrained network, ResNet-34 (He et al., 2016), for regressing camera pose. It further adopted encoder-decoder design and used skip connection to move the features from the early layers to the output layers. This further enhances the accuracy to 0.23 m. Despite the good performance achieved, all the networks contain a large number of neurons and rely on pretrained networks as the training of such network will otherwise become too time consuming to be practical.

To enhance the training efficiency of CNN, we, for the first time according to the authors' knowledge, use an image descriptor to reduce the input dimension of CNN by almost two orders of magnitude. Subsequently the number of neurons required for CNN are significantly reduced, leading to highly efficient training compared to previously reported CNN models without sacrificing the accuracy.

## 2 MODEL

In this section, we apply our network to image based indoor localization. The task is that given an image *I* taken by a camera, find the global Cartesian coordinates [x,y,z] of the camera location **P**.

As shown in Fig. 1, instead of directly feeding images to the CNN, our model added a SURF descriptor to extract a set of $64$ dimensional features from the image. Since the number of features of each image is different, we choose $N$ features with the highest Hessian threshold (Bay et al., 2008) from each image as the input of CNN. The CNN consists of a typical 5-layer CNN with max pooling and batch normalization. Its output is flattened to a fully connected layer (Ioffe & Szegedy, 2015). The output layer comes with 3 neurons for the position of [x,y,z]. Here, we used the mean square error as our loss function.

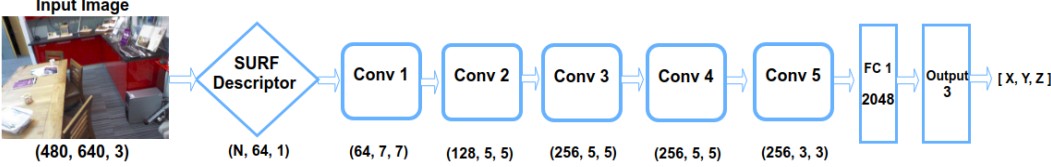

Figure 1: The architecture of SurfCNN. (N: Number of SURF Features, Conv: Convolutional Layer and FC: Fully Connected Layer). The number of filers and width of each convolutional layer are listed in the bracket beneath the corresponding blocks.

## 3 PERFORMANCE ANALYSIS

To validate our model, we selected 2 representative scenes from the 7-scene dataset Glocker et al. (2013). Specifically, we choose the images with $480 \times 640 \times 3$ pixels from the scene of "chess" which is rich in features and "stairs" that has relative simple structure. The accuracy of the estimated position where the camera took these images is used as the figure of merit to compare our model with published results.

To determine the number of SURF features needed as CNN input, we plot the localization accuracies of both scenes as a function of number of features selected in Fig. 2. As a comparison, the accuracies of PosNet(red dotted line, (Kendall et al., 2015)), Pose-LSTM (green dashed line, (Hazirbas et al.)) and Pose-Hourglass (yellow dash-dotted line, (Melekhov et al., 2017)) are shown on the same sub-

plots. In the scene of chess, our SurfCNN outperformed PoseNet with as few as 5 features. It further reached an accuracy better than Pose-LSTM when adopting 200 features. It is worth mentioning that due to the large feature size associated with the images of chess, selecting only 300 features is insufficient to beat the Pose-Hourglass model. On the other hand, as shown in Fig. 2, in the relative simpler scene of stairs, the SurfCNN reached the same accuracy as Pose-Hourglass by selecting 300 features.

The advantage of SurfCNN is evident in Tab. 1. Assuming 300 SURF features are chosen, the input dimension of CNN is reduced to $19,200$ from $921,600$. As a result of this $48$ fold input reduction, SurfCNN only needs 7 network layers and does not require a pretrained network. In contrast, PoseNet, Pose-LSTM and Pose-Hourglass consist of 24, 28 and 35 layers and all require additional pretrained network. Note that the number of parameters (excluding pretrained network) in Pose-Hourglass is almost doubled to SurfCNN while with sufficient number of features retained SurfCNN can reach the same accuracy. Finally, SurfCNN typically takes only around 1.5 hour for training while the training of pretrained networks in other models alone will take days or even longer.

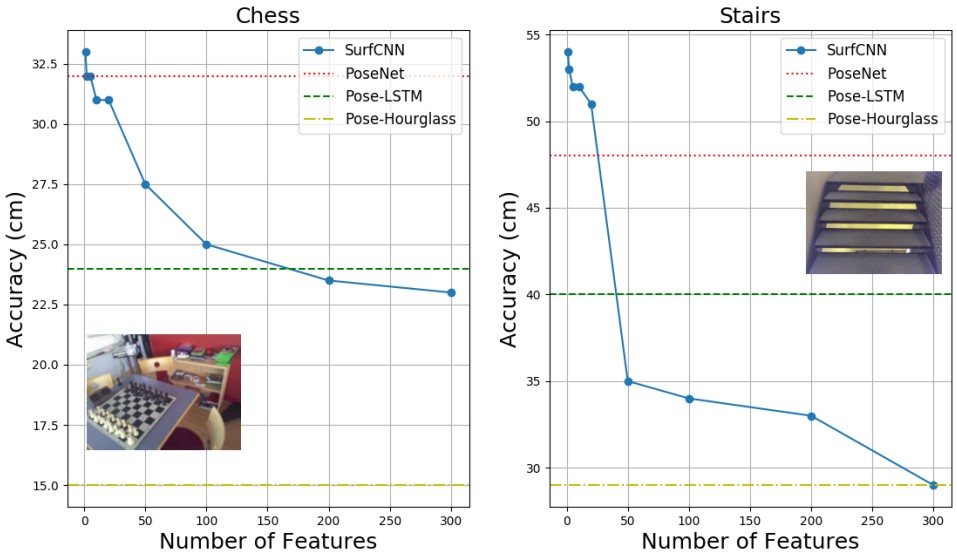

Figure 2: The effect of varying the number of features on accuracy of SurfCNN compared to Posenet (Kendall et al., 2015), Pose-LSTM (Hazirbas et al.) and Pose-Hourglass (Melekhov et al., 2017)

Table 1: The comparison of SurfCNN, Posenet (Kendall et al., 2015), Pose-LSTM (Hazirbas et al.) and Pose-Hourglass (Melekhov et al., 2017)

| Network | Layers | Pretrained Network | Pretrained Parameters | Total Parameters | Accuracy (cm) | |
| --- | --- | --- | --- | --- | --- | --- |
| | | | | | Chess | Stairs |
| SurfCNN | 7 | None | 0 | $1.3 \times 10^7$ | 23 | 29 |
| PoseNet | 24 | GoogLeNet | $1.1 \times 10^7$ | $2.35 \times 10^7$ | 32 | 48 |
| Pose-LSTM | 28 | GoogLeNet | $1.1 \times 10^7$ | $2.15 \times 10^7$ | 24 | 40 |
| Pose-Hourglass | 35 | ResNet-34 | $2.3 \times 10^7$ | $4.5 \times 10^7$ | 15 | 29 |

## 4 CONCLUSION

In conclusion, we implemented SurfCNN that used a SURF descriptor to reduce the input dimension of CNN. Benefited from the advantages of both, our network outperformed PoseNet and Pose-LSTM without the need for a pretrained network. With the sufficient number of features, SurfCNN reaches the same accuracy as Pose-Hourglass with only half the parameters even excluding the pretrained network. This advantage is essential in realtime localization tasks where memory size is small. This approach is not only efficient, but also versatile to all other CNN related applications to make the training highly efficient.

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

## ACKNOWLEDGEMENT

The research is sponsored by Fortinet Inc.

