# OpenReview forum: "SURFCNN: A DESCRIPTOR ENHANCED CONVOLUTIONAL NEURAL NETWORK"
_ICLR.cc/2018/Workshop — Reject_

### Official Review · AnonReviewer2 · 2018-03-05
**This paper proposes SURF descriptor as input of CNN to reduce the input dimensions of CNN**

**Rating:** 6
**Confidence:** 4

**Review:**

This paper showed that using the SURF descriptor as input of CNN is possible to achieve as similar performance as the methods which use pre-trained CNN as input in an indoor localization application. Although it is unclear if this approach generality works well for other tasks with the limited experiment of this paper, the method have a merit for reduction of the parameters at least in this task.

Pros
- The proposed method reduce the total parameters of the CNN.
 -The proposed method does not require pre-training of CNN.

Cons
- It is no reasoning to conduct 2D convolution operation for sampled SURF descriptors because the SURF descriptors are vectors, not a feature map.
- The performance of Chess scene is lower than Pose-Hourglass.
- Only two scenes from 7-secene datasets and only the accuracy of x,y,z position were evaluated although the camera orientation was also evaluated on papers of compared methods.

---

### Official Review · AnonReviewer1 · 2018-03-10
**unclear**

**Rating:** 4
**Confidence:** 4

**Review:**

The paper proposes to learn pose estimation from SURF descriptors instead of pixels and the results suggest that this makes learning easier and faster without sacrificing accuracy. This is all fine. The problem is that it is unclear what exactly the model is doing. It seems like there is a set of 300 SURF descriptors per image, but how are these fed into a convnet that assumes a regular input grid ? What is the 7x7 filter at the first layer doing with the bag of descriptors ? There seems to be some problem with the exposition.

---

### Decision · Program_Chairs · 2018-03-20
**ICLR 2018 Workshop Acceptance Decision**

**Decision:**

Reject

**Comment:**

Based on the reviews, this paper has not been accepted for presentation at the ICLR workshop. However, the conversation and updates can continue to appear here on OpenReview.